# Higher fall rates and broader kinematic diversity in bilateral versus unilateral unconstrained slips

**Abderrahman Ouattas**[1]*, **Andrew Walski**[1], **Seongwoo Mun**[1], **Corbin M. Rasmussen**[1,2], **Nikolaos Stergiou**[1,3], **Nathaniel H. Hunt**[1]*

**1** Division of Biomechanics and Research Development, Center for Research in Human Movement Variability, Department of Biomechanics, University of Nebraska at Omaha, Omaha, Nebraska, United States of America, **2** Department of Exercise Science and Pre-Health Professions, College of Arts and Sciences, Creighton University, Omaha, Nebraska, United States of America, **3** Department of Physical Education and Sport Science, Aristotle University, Thessaloniki, Greece

\* aouattas@unomaha.edu (AO); nhunt@unomaha.edu (NHH)

## Abstract

Identifying and categorizing slip types by their risk level is essential for developing effective training protocols to prevent slip-induced falls. While previous research has thoroughly examined various slip types in sagittally constrained and unconstrained unilateral slips, little is known about the broader range of unconstrained unilateral or bilateral slips that closely mirror real-world conditions. In this study, we addressed three primary questions: (1) Do bilateral slips produce higher fall rates than unilateral slips? (2) Do bilateral slips exhibit greater slip diversity? (3) Does separating diverse slips by slip type reveal differences in severity, such as whole-body angular momentum and foot velocities? We administered three sudden, unconstrained unilateral or bilateral slips, using a wearable perturbation device, during over-ground walking to 20 younger adults (age: 27±4.71 years in the unilateral group, 26.5±4.03 years in the bilateral group), with whole-body kinematics and ground reaction forces recorded. Probabilistic Graphical Models assessed slip diversity and severity for each group, while multivariate general linear models examined whole-body angular momentum and average foot velocities after trailing leg touchdown across slip types. Six falls (21.42% fall rate) occurred in the bilateral slip group, whereas no falls were observed in the unilateral group, indicating that bilateral slips are more severe. Entropy analyses showed that bilateral slips had 21.44% greater outcome diversity compared to unilateral slips. We identified three slip types, high, medium, and low severity, based on frontal and sagittal foot movements following trailing foot touchdown. These categories differed significantly in whole-body angular momentum and foot velocities, suggesting distinct severities and underlying biomechanics. Future studies should examine whether different slip contexts (unilateral vs. bilateral) and slip types demand unique coordination strategies for fall prevention.

**Data availability statement:** The raw unidentified data supporting the conclusions of this article has been added as supplementary files. Other data that include personal identification numbers and other personal information cannot be shared publicly as it would breach compliance with the protocol approved by the research ethics board.

**Funding:** This work was supported by the National Institutes of Health; NIH 2P20 GM109090-06, NIH R15AG063106-01, and NIH NIGMS P20GM152301. The funders had no role in study design, data collection and analysis, decision to publish, or preparation of the manuscript.

**Competing interests:** The authors declare that the research was conducted in the absence of any commercial or financial relationships that could be construed as a potential conflict of interest.

## Introduction

Falls and slips constitute a major and growing public health concern. In 2020, the Centers for Disease Control and Prevention reported that falls caused over 40,000 deaths, making them the third leading cause of avoidable injury-related mortality [1]. More specifically, slips accounted for nearly 40% of all falls, and slip-induced falls on the same level contributed to 50,100 injuries [2]. Despite these large numbers, understanding the nature of slip-induced falls is limited. One study that categorized different slip types found that about half of observed slip-induced falls during mail delivery were linked to diverse surface conditions (e.g., ice, snow, wet or loose surfaces, grass, leaves) [3]. Unlike trips where previous research has identified trunk flexion as a causal biomarker to achieve recovery [4] due to the fact that a trip causes the trunk and center of mass (CoM) to fall forward, slips on the other hand cause the foot to slide in multiple directions [3]. We hypothesize that the latter yields a diverse set of multi-dimensional behavioral outcomes that makes slips especially difficult to recover from. Over the past two decades, non-fatal injuries due to slip-induced falls increased by 160% [5], while fatal injuries increased by 300% [6], which could be caused by the diversity and severity of slips that we lack awareness of; consequently, we lack awareness of the specific countermeasures to avoid slip-induced falls.

A principal challenge in recovering from a sudden, unexpected slip arises from the varied paths the slipping foot can take in the horizontal (shear) plane, specifically in anterior–posterior and medial–lateral directions. Each slip can generate distinct foot trajectories and corresponding upper-body movements, potentially disrupting balance in multiple planes. Bilateral slips that are unconstrained, meaning friction is sufficiently low that both feet can slide freely in the horizontal plane, can further intensify the variety and severity of these outcomes [7]. Indeed, our previous work [7] showed that such unconstrained bilateral slips lead to diverse kinematics across the lower and upper body, with frontal-plane feet velocities relative to the CoM providing the best discrimination between falls and recoveries. Although the slipping foot itself remains in the horizontal plane, these perturbations can induce full-body angular motions in the frontal, sagittal, or even transverse planes, posing an especially difficult challenge for balance recovery. While most laboratory studies constrain slips to the sagittal plane for experimental control and repeatability, slips in everyday environments are typically unconstrained because low-friction surfaces permit foot motion in any horizontal direction. Epidemiological evidence from community-dwelling older adults shows that 59% of all recorded falls are slips or trips, many involving lateral motion and sideways impacts [5]. In a study of ambulatory older women, explicit lateral motion occurred in roughly one-third of documented falls [8]. Experimental studies that remove directional constraints confirm this pattern: unconstrained slips often exhibit substantial mediolateral foot motion, especially when slips occur during curvilinear gait [9,10]. Real-life video analyses likewise report frequent lateral stepping responses, with compensatory steps observed in 76% of long-term-care falls and most prevalent in sideways falls [11]. Together, these findings indicate that the unconstrained slips studied here realistically represent the multidirectional conditions commonly encountered in daily life.

Slip severity offers a useful basis for distinguishing slipping movements that end in recoveries from those that end in falls, but this approach becomes less reliable for a wide range of slip types. In most laboratory studies, slip severity is defined by thresholds of peak slip distance or velocity, typically measured in a single plane of motion (often the sagittal) to classify outcomes as falls or recoveries [12]. However, Cowin and colleagues [13] demonstrated that "execution diversity" (i.e., trial-to-trial variability) must be evaluated within the same strategy type; comparing data across different strategies makes defining a single slip-severity threshold both difficult and potentially misleading. For unconstrained slips, where movement can manifest in diverse biomechanical patterns, grouping all trials together oversimplifies a set of processes with a highly varied probability distribution. Such heterogeneous data are rarely independent and identically distributed, so conventional dispersion metrics (e.g., standard deviation or interquartile range) may fail to capture the true extent of slip diversity.

Laboratory studies of sagittally constrained unilateral slips at early stance have traditionally recognized three main slip types: (1) feet-forward falls, in which the trailing limb lands in front of the slipping leg and both feet slide forward; (2) split falls, in which the trailing limb lands behind the slipping leg; and (3) aborted-step recoveries, where individuals offload the slipping leg's ground reaction forces (GRFs) and bear more weight on the trailing, non-slipping leg [14–27]. More recent work has further detailed constrained slip types, including (1) backward balance loss, in which the CoM travels beyond the posterior stability boundary; (2) no backward balance loss, in which the CoM remains within anterior–posterior limits; (3) limb collapse, in which the trailing limb lands behind the CoM, potentially triggering a sudden drop; and (4) instability, in which the trailing limb lands ahead of the CoM, creating greater instability [28,29].

In contrast, two recent studies have categorized slip types under unconstrained conditions. Hu et al. (2022) [30] found that larger sagittal distances between the trailing and leading legs, along with higher and later slipping-foot velocity, reduced recovery success. Similarly, Allin and colleagues (2018) [31] identified four categories of unconstrained unilateral slips, recoveries, feet-split falls, feet-forward falls, and lateral falls, and reported that a combination of slipping-foot velocity, relative positioning of the trailing and leading feet at touchdown, and frontal-plane speeds during the first 50 ms accurately distinguished slips that ended in falls from those that did not. Other experiments with unconstrained unilateral slips further link slower frontal-plane velocities, increased trunk flexion, and faster shoulder extension with improved recovery [4,9,31–47]. Despite this substantial body of work, fundamental gaps remain regarding slip type classification when individuals encounter a wider array of unconstrained unilateral or bilateral slips.

Classifying unconstrained slip types serves two primary purposes: first, to distinguish high- from low-severity slips based on whether they result in a fall or a recovery, and second, to determine whether each slip type constitutes a distinct motor task requiring specialized balance-restoring responses. Before implementing effective slip-training protocols to reduce falls, it is essential to identify the full range of slip types that can arise under unconstrained unilateral and bilateral conditions, as well as to categorize these types by risk level. Accordingly, this study addressed three main questions: (1) Do bilateral slips lead to higher fall rates than unilateral slips? (2) Do bilateral slips produce greater slip diversity than unilateral slips? (3) Does disaggregating diverse slipping trials by slip type reveal meaningful differences in slip severity (i.e., whole-body angular momentum, feet velocities)? We identified whole- body angular momentum as a slip severity outcome measure because unilateral unconstrained slip literature identified it as a significant predictor of the outcome (fall/recovery) and severity (severe/mild) of the slip [48]. Foot velocity was used as a slip severity outcome measure because we previously showed that frontal plane feet velocities were the best discriminators of falls and recoveries during unconstrained bilateral slips [7]. We hypothesized (H1) that simultaneous, unconstrained bilateral slips would exhibit greater slip diversity than unilateral slips, (H2) that they would pose a higher fall risk, and (H3) that slip types would differ in whole-body angular momentum and feet velocities.

## Materials and methods

### Participants

Twenty young adults with no prior experience of in-lab slips participated in this study. They were randomly assigned in equal numbers to a unilateral or bilateral slip group (Table 1). At enrollment, all participants self-reported the absence of

**Table 1. Participant demographics (Mean (Standard Deviation)).**

| Participants | Age (years) | Height (m) | Weight (kg) | BMI (kg/m²) | Sex |
|---|---|---|---|---|---|
| **Unilateral Slips (n = 10)** | 27 (4.71) | 1.73 (0.10) | 73.70 (8.27) | 24.66 (3.16) | 2F |
| **Bilateral Slips (n = 10)** | 26.5 (4.03) | 1.71 (0.11) | 69.26 (14.71) | 23.56 (5.27) | 2F |

any exclusion criteria: (1) previous participation in a slip study or prior in-lab slip exposure, (2) uncontrolled hypertension, (3) peripheral arterial disease, (4) vertigo, (5) Meniere's disease, (6) chronic dizziness, (7) back injury history, (8) surgery affecting mobility, (9) any neurological disorder that limits walking (e.g., stroke, Parkinson's disease, multiple sclerosis), or (10) muscle weakness disorders (e.g., muscular dystrophy, myopathy, myositis). The recruitment period for this study began on August 9th, 2021, and concluded on March 23rd, 2022. Written informed consent, approved by the University of Nebraska Medical Center's Institutional Review Board (IRB# 0181-21-EP), was obtained before any study procedures began.

## Experimental protocol

Upon arriving at the Biomechanics Research Building at the University of Nebraska at Omaha and providing written informed consent, each participant had their height and weight measured. They also reported any history of falls prior to the study date, specifying the most recent fall in daily activities (excluding sporting events) and its cause (S1 Dataset). The protocol is outlined in Fig 1. A and all experimental procedures took place in the main Motion Analysis Laboratory (Fig 1B and 1E). After collecting anthropometric data, participants completed the Y-Balance Test (YBT) [49–52] before starting the slip trials (Fig 1A and 2A–C). The YBT, which has no ceiling effects, was used to assess and compare baseline balance abilities across groups.

Following the YBT, participants donned a form-fitting suit and were fitted with a ceiling-mounted safety harness (Fig 1B and 1E). They briefly sat in the harness to ensure that, in the event of a slip, their knees would remain off the ground. Participants then walked back and forth across a 10-meter walkway (Fig 1B), during which all trials were captured on video using a Canon VIXIA HF R82 at 30 Hz (Canon Inc., Ota City, Tokyo, Japan). Each participant also signed a video and photo release form prior to recording.

Our in-lab–manufactured Wearable Apparatus for Slipping Perturbations (WASP; Fig 1E) was used to create diverse unconstrained slipping conditions. The latest version of this device is described in detail elsewhere [7]. Briefly, the WASP was designed to be worn over any standard athletic shoes through its polyethylene insole sock, covered by a second layer of polyethylene outsole sock, with a lubricated environment in between. When the WASP triggers wirelessly by the experimenter, it immediately reduces friction by separating the outsole and insole socks within 50 milliseconds to allow the lubricated insole to slide over the lubricated outsole and continue to slide over the floor. We previously showed evidence that the WASP reduces the dynamic coefficient of friction from approximately m = 0.6 ± 0.1 (outsole and floor) to a coefficient close to that experienced between shoes and ice (insole to outsole m = 0.14 ± 0.04, insole to floor m = 0.12 ± 0.0), and lower than the minimum available dynamic coefficient friction (m = 0.2) during walking [7]. Such conditions cause unexpected and unconstrained slipping conditions that are close to real-world slips [7,9,10,44]. Worn athletic shoes were normalized across all participants. Participants rested for about two minutes between each trial to allow for WASP resetting (Fig 1A). Slips were induced either unilaterally (Fig 1B and 1C) or bilaterally (Fig 1E and 1F) at a random time within the first five minutes of each trial. All slips were triggered by the experimenter at heel strike and initiated during early stance of the dominant limb due to any human delay and error (0–33.3% of the stance phase; Fig 1C and 1F). Several previous research studies have identified 0–33% of the stance phase as the load absorption phase using shear ground reaction force data in healthy and diseased populations [53,54]. Trials that did not result in slip initiation at the load absorption phase were excluded from the analyses. In unilateral slips, only the WASP worn on the dominant limb was triggered (Fig 1C). In bilateral slips both WASPs activated simultaneously (i.e., at the same exact timepoint) when the dominant foot

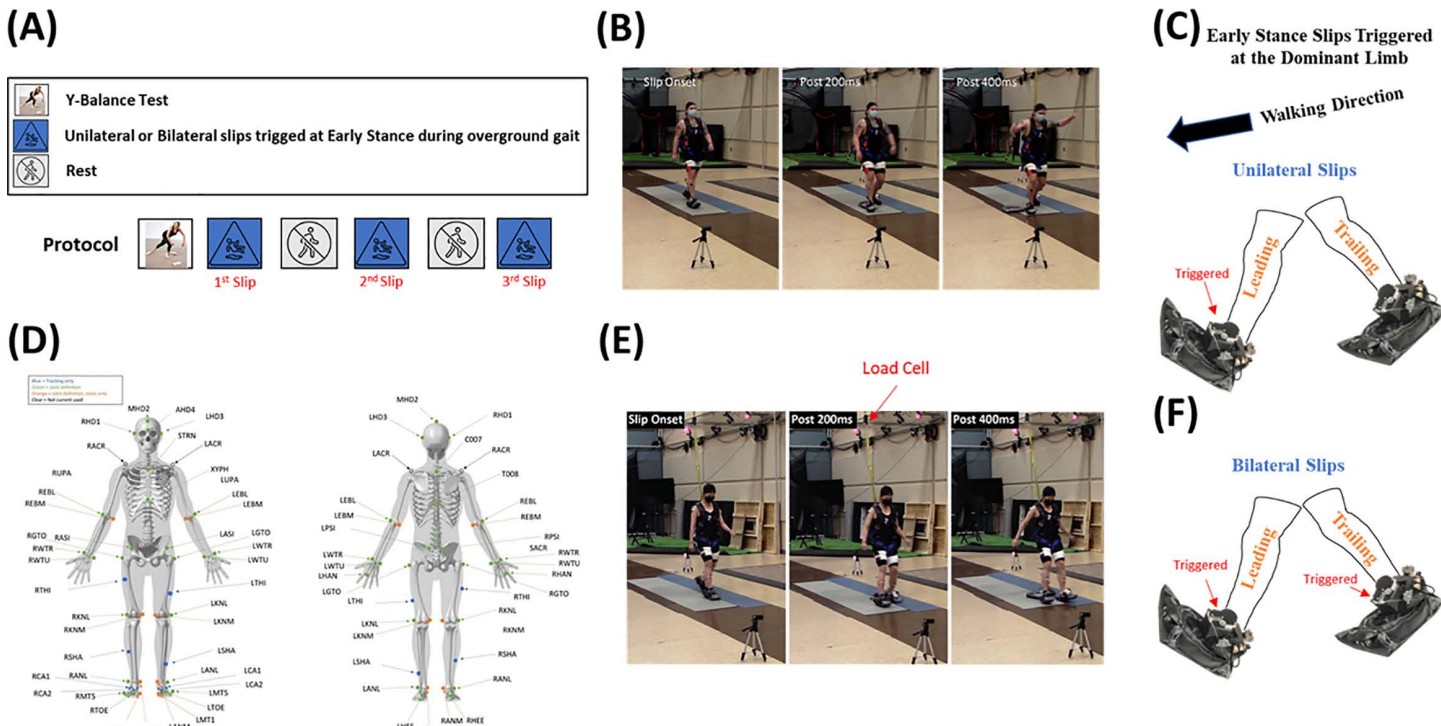

**Fig 1. Experimental protocol.** (A) Protocol Outline. Participants completed the YBT followed by three unilateral or bilateral slips induced at their dominant foot's early stance. Participants rested for approximately two minutes while the researcher reset the Wearable Apparatus for Slipping Perturbations (WASP). (B) A panorama demonstration of a unilateral slip administered at the dominant foot (right side in this case), indicated at slip onset, 200ms, and 400ms post slip onset. (C) A stick figure demonstration of the differences between unilateral and bilateral slips. In this example, the right foot is the dominant and leading foot, while the left foot is the non-dominant and trailing foot. The WASP is always worn on both feet. Unilateral slips only trigger the WASP worn on the dominant foot, while bilateral slips trigger both WASPs simultaneously (D) Retroreflective marker-set used in this study to record full-body kinematics (E) A panorama demonstration of a bilateral slip administered at the dominant foot (right side in this case), indicated at slip onset, 200ms, and 400ms post slip onset. The figure also shows the load cell attached to the harness used to quantify falls and recoveries. (F) A figure demonstration of the WASP trigger during bilateral slips. In this example, the right foot is the dominant and leading foot, while the left foot is the non-dominant and trailing foot. The WASP was always fitted on both feet. bilateral slips triggered both WASPs simultaneously.

entered early stance (Fig 1F), initiating an early-stance slip on the dominant (leading) foot and a late-stance slip (66.8–100% of stance) on the non-dominant (trailing) foot prior to swing phase. Each WASP was triggered remotely and manually by the same researcher to reduce inter-assessor variability since WASPs were not manufactured with an automated feature that detects gait events during the stance phase. Because all WASPs connect to the same device through Bluetooth and can only be triggered manually and simultaneously by the same researcher, bilateral slips were triggered when the dominant foot reached early stance, which automatically triggered the non-dominant foot's WASP during late stance. Slips were initiated at the non-dominant's foot late stance prior to swing phase and the foot continued to slip after touchdown. Participants were instructed to try their best not to fall upon slip initiation as if they were experiencing a slip outdoors until either equilibrium or a fall is reached and then to stop walking. Upon concluding each trial, researchers walked to each participant and assisted them to carefully walk back to the start line, unhooked the harness for them to sit down and rest on a chair, reset the WASP, and cleaned the floor.

Kinetic and kinematic data (Fig 1B and 1C) were recorded during each trial. Because gait speed can influence slip-proactive control [37], both groups were asked to walk at a fixed speed of 1.3±0.1 m/s, a comfortable pace for men and women [55]. A Dashr system (Dashr, Lincoln, NE, USA) measured overground walking speed, and participants received verbal feedback to slow down or speed up if they fell outside the target range.

## Biomechanical analyses

A set of 56 retroreflective markers was placed on the feet, shanks, thighs, pelvis, torso, hands, forearms, upper arms, and head to capture whole-body kinematics (Fig 1D). Kinematics were recorded with a 20-camera high-speed infrared motion capture system (Motion Analysis Corp., Santa Rosa, CA, USA), which sampled 3D marker positions at 102.4 Hz. All kinematic data were interpolated (maximum gap of 10 frames, polynomial order of 3) and low-pass filtered at 10 Hz using a fourth-order Butterworth filter. Kinetics were collected via a harness-mounted load cell with a maximum capacity of 500 kg (HT Sensor Technology Co. LTD, Xi'an, China), sampled at 80 Hz, and via eight floor-embedded force plates (5 Optima and 3 Gen-5 force plates – Advanced Mechanical Technology, Inc., Watertown, MA), sampled at 1,024 Hz. Data were processed in Cortex 8.1 software (Rohnert Park, CA, USA), and subsequent biomechanical analyses were performed in custom Visual3D pipelines (C-Motion Inc., Germantown, MD, USA).

All biomechanical data were evaluated during the "slipping timeline." This interval began when the dominant foot started sliding and ended once the sum of both feet's absolute velocities reached zero. Slip initiation was visually confirmed in Visual3D after the WASP was triggered: if a foot's displacement exceeded 30 mm [34], the video was traced backward frame by frame until the foot no longer slid. The very next frame was designated as the "slip start" (S1 Table). Slip onset was chosen as the start of the slipping timeline instead of heel strike to normalize all trials to the same slip start and ensure that the analyzed biomechanical data used to quantify slip severity and diversity are data extracted during the actual slip; thus, the slipping timeline has been identified when the dominant foot start sliding until both feet reached a complete stop.

## Qualitative assessment

A frame-by-frame qualitative biomechanical analysis was conducted in Visual3D for each slip trial and cross-referenced with the real-time videos. Two bilateral trials were excluded because they did not induce a proper early-stance slip at the dominant foot (one due to a WASP malfunction and one due to a late trigger). This resulted in a final sample of 28 bilateral slip trials and 30 unilateral slip trials.

These qualitative assessments were undertaken to characterize both unilateral and bilateral slip contexts at WASP trigger, followed by their severity and outcome using PGMs (S1 Table). Trailing foot touchdown was confirmed by changes in the center of pressure (CoP) and GRFs. Slip outcomes were classified into (1) complete recoveries, (2) harness-assisted recoveries, or (3) falls (defined in Section 3.2). Slip types were then categorized according to whether the leading and trailing feet, relative to the CoM, were arrested or continued moving in the frontal and/or sagittal planes after trailing foot touchdown. Foot position and velocity relative to the CoM were chosen to identify slip types because the change in whole-body angular momentum depends on the magnitude of the resultant GRFs and the location of the base of support relative to the CoM [41]. This implies that placing the foot behind the CoM and applying GRFs yield a forward whole-body angular momentum, and the opposite is true, while placing the foot lateral to the CoM and applying GRFs yield a contralateral whole-body angular momentum, and the opposite is true.

All foot velocities were inspected within 30 ms of slip initiation and trailing foot touchdown in Visual3D to determine whether the foot was still or in motion before H-reflex latency could occur (~29.8 ± 2.74 ms in healthy adults) [56]. We chose to initially inspect foot velocities within 30 msec of slip initiation and trailing foot touchdown to quantify the velocity of the foot solely due to the reduced frictional demands caused by the slips before any human reflexes interferes, which could increase or decrease in foot velocity and alters the actual velocity of the foot caused solely by the slip. Average foot velocity (Equation 1) was extracted during the 100 ms following trailing foot touchdown because foot behavior immediately after touchdown is critical to differentiating falls from recoveries. This time window both precedes the typical onset of voluntary muscle activation (~180 ms in take-off/landing activities) [57] and occurs before the fall threshold is reached in all fall trials; thus, is the only timeline that can be normalized to all trials (recovery, harness-assisted recovery, and falls) and provides accurate foot velocity data that are not caused by voluntary muscle reaction post trailing foot touchdown.

## Quantitative Assessment

Falls were determined if the peak tensile load applied to the harness exceeds 30% of the participants' bodyweight [58] (S1 Table). Recoveries were identified if the tensile load applied to the harness did not exceed 4.5% for at least 1 second on average. Harness-assisted recoveries were identified when individuals applied more than 10.28% of body weight of tensile load into the harness across 0.2 seconds with a peak less than 30% of body weight. Tensile loads were measured using a load cell connected to the harness. Kinematics were applied to identify the different slip types using leading (dominant) and trailing (non-dominant) feet velocities within 100ms of trailing foot touchdown (Equation 1),

$$\overline{v} = \frac{\Delta \overrightarrow{x}}{100ms} \tag{1}$$

where $\overline{v}$ is average velocity and $\Delta \overrightarrow{x}$ is the displacement of each foot. Kinematics were also used to identify the different slip states by measuring feet velocities at slip initiation and trialing foot position relative to the CoM at trailing foot touchdown (S1 Table). The foot was labeled arrested with a recorded velocity of ≤0.099 m·s$^{-1}$ or in motion with a recorded velocity of ≥0.100 m·s$^{-1}$.

Whole-body angular momentum about the body's CoM (L) was calculated (Equation 2) in all planes of motions (frontal, sagittal, and transverse) and was extracted at three timepoints: 1) slip initiation, 2) trailing foot touchdown, 3) 100ms post trailing foot touchdown.

$$L = \sum_{n=1}^{N} m_n \left( r_{CoM_n} \times v_{CoM_n} \right) + I_n \cdot \omega_n \tag{2}$$

In this calculation of L, the body CoM is considered the origin. The whole-body CoM position was derived from the weighted mean position of all individual segment CoM in our 15-segment model (2 feet, 2 shanks, 2 thighs, 1 pelvis, 1 trunk, 1 head, 2 upper arms, 2 forearms, 2 hands). The bold symbol r represents the displacement vector from the body CoM to the segment CoM. The bold symbol v represents the velocity vector of the segment CoM with respect to the body CoM. The symbol $m_n$ is the scalar mass of the n$^{th}$ segment. $I_n$ and $\omega_n$ are the moment of inertia and angular velocity of n$^{th}$ segment about its own CoM, respectively. Segment angular velocities ($\omega_n$) were calculated as the derivative of the segment's angle. All segment parameter values, including $m_n$, were estimated from the whole body mass of each individual using predefined methods reported by Dempster [59]. $m_n$ has been normalized to each participant's mass. The moments of inertia $I_n$ and CoM for each segment were calculated using validated experimental methods by Hanavan [60]. L directions are defined as follows; Y=Anteroposterior (frontal plane), X=Mediolateral (sagittal plane), Z=vertical (transverse plane). L was calculated as the sum of angular momentum for all 15 body segments.

## Statistical analyses

The independent t-test was used to assess the statistical differences of the YBT composite score between unilateral and bilateral slip groups for both the dominant and non-dominant legs. The YBT composite score was calculated based on Equation 3.

$$Y - Balance\ Composite\ Score = \frac{Sum\ of\ the\ Greatest\ Reach\ in\ each\ Direction}{3\ \times\ Limb\ Length} \times 100 \tag{3}$$

To describe the qualitative biomechanical analyses aspect, we modeled the slipping process as a stochastic process by creating two probabilistic graphical models (PGMs): one for unilateral slips, $Y_1$, and one for bilateral slips, $Y_2$. Each state in the PGM represents each distinct slipping event as a discrete state, and the transition from one slipping event to the next

as a set of transition probabilities (S1 Table). To assess the diversity of each slip type we calculated the expected surprise or entropy ($H$) of each PGM (Equation 4),

$$H(Y) = E\left(I(X)\right) = \sum_{ij} p(x_i)P_{ij} \times log_2 \frac{1}{P_{ij}}$$

(4)

where $E\left(I(X)\right)$ is the expected surprise of a sequence of slipping events, $X$. The probability of a slipping event, $p(x_i)$, is the sum of the probability of all paths to $x_i$. $P_{ij}$, is the probability of a transition from $x_i$ to $x_j$. Each element of the sum in the above equation can be considered in two parts, where the first part is a conditional probability, $[p(x_i)P_{ij}]$, the probability that a given transition will occur, $P_{ij}$, given that the state $x_i$ has occurred. The second part represents the uncertainty as a quantity of information in bits, $[log_2 \frac{1}{P_{ij}}]$. We selected PGMs because slips unfold as an ordered sequence of states that explicitly mirrors the slip timeline, such as heel contact, slip onset, and trailing foot touchdown. PGMs can explicitly capture that structure while remaining data efficient. Also, compared with black-box machine learning classifiers, PGMs (i) embed domain knowledge directly in the graph, (ii) propagate uncertainty to yield interpretable, probabilistic fall risk estimates, and (iii) provide an intrinsic entropy output that we use as a slip diversity metric. These capabilities are particularly useful when working with modest sample sizes and when clinical interpretability is a priority [61].

Multivariate general linear models (GLMs) [2×3×3; Slip Group (Unilateral vs Bilateral) × Slip Timepoint (Slip initiation vs trailing foot touchdown vs 100ms post-trailing foot touchdown) × Slip Type (Type I vs Type II vs Type III)] were used to quantitatively analyze slip severity by assessing the differences in L across slip types and slip groups in all three planes of motion. Additionally, GLMs [2×3; Slip Group (Unilateral vs Bilateral) Slip Type (Type I vs Type II vs Type III)] were used to quantitatively analyze slip severity by assessing the differences in average feet velocities from trailing foot touchdown until 100ms post trailing foot touchdown across slip types and slip groups in the frontal and sagittal planes of motion. GLMs were used instead of ANOVA, MANOVA, and traditional logistic regression models because they offer several advantages; 1. Does not require transformation of response Y to achieve a normal distribution 2. Provides model autocorrelation for repeated measures, 3. Accounts for missing data, 4. Does not require variances of the independent variable to be equal, 5. Allows a customized model according to the nature of the dependent variable, 6. Is not limited to fixed repeated time intervals, and 7. Less prone to type II errors [62,63]. Least Significant Difference (LSD) was used to compare main effects. Cohen's d was used to calculate effect size for the independent t-test and partial eta square ($\eta^2$) was used to calculate effect sizes for GLMs ($\eta^2 = \frac{Treatment\ Sum\ of\ Squares}{Total\ Sum\ of\ Squares}$) [64]. Custom MATLAB codes (MathWorks, R2022a, Natick, MA, USA) and SPSS (IBM, v.28, Armonk, NY, USA) were used for statistical analyses and data visualization, and the significance level was set at p = 0.05.

## Results

### YBT

Participants within the unilateral- and bilateral slip groups showed similar YBT composite scores for both the dominant (Unilateral: 113.61 ± 13.09, Bilateral: 113.11 ± 13.05) and non-dominant (Unilateral: 115.50 ± 12.73, Bilateral: 116.40 ± 9.72) legs (Fig 2). T-test showed no significant difference between the unilateral and bilateral slip groups for both the dominant (t(18) = 0.086, p = 0.932) and non-dominant (t(18) = −0.177, p = 0.862) legs. Cohen's d effect sizes showed small and moderate effects for the dominant (Cohen's d = 0.38) and non-dominant (Cohen's d = − 0.79) sides, respectively (Fig 2).

### Unilateral vs bilateral slips' diversity and severity

Prior fall incidence did not influence fall rates under either unconstrained unilateral or simultaneous bilateral slip conditions (S1 Dataset). Given the probabilistic nature of our PGMs and this study's objectives, we used PGMs to (1) compare slip

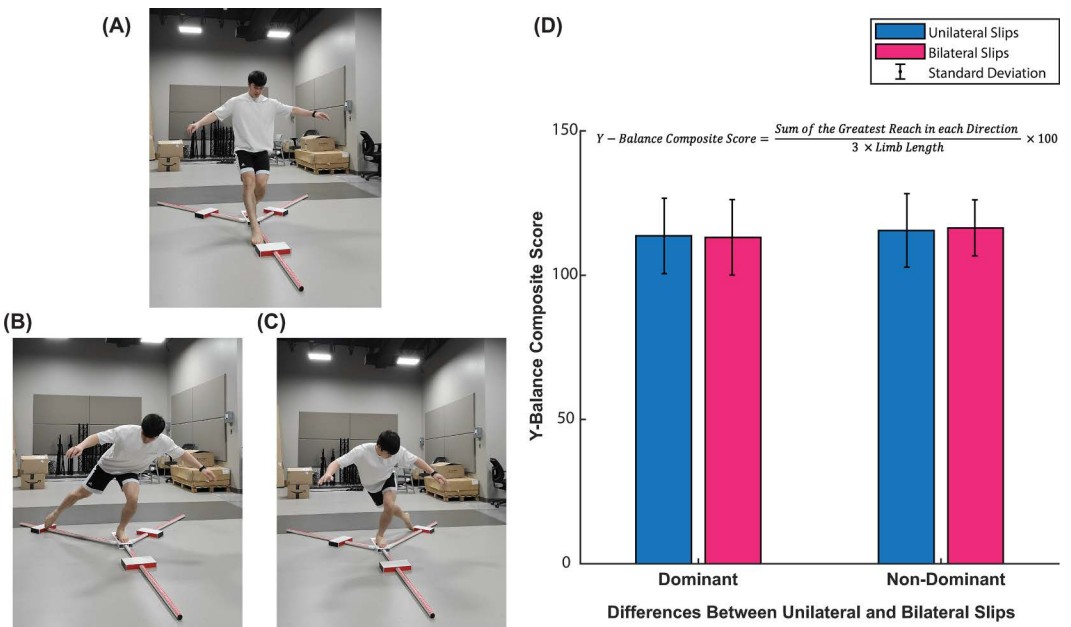

**Fig 2. Y-balance test findings.** (A), Figure Demonstration of the YBT Anterior Reach for the left leg (non-dominant in this case). (B), Figure Demonstration of the YBT Posterolateral Reach for the left leg. (C), Figure Demonstration of the YBT Posteromedial Reach for the left leg. (D), Differences in YBT Composite score for each leg (dominant and non-dominant) color coded by slip type (unilateral vs bilateral slips). Results show no significant statistical differences between leg dominance and slip type. Standard deviation is represented as error bars.

diversity and severity between unilateral and bilateral conditions and (2) categorize slip types based on early-stage foot velocities/positions and slip outcomes. On average, trailing leg touchdown occurred $255 \pm 91$ ms after slip initiation. During fall trials, participants reached the harness load threshold of 30% body weight at $515 \pm 175$ ms, allowing time to respond to the initial slip and trailing leg touchdown. The entire slipping timeline (from slip start until both feet came to rest) lasted about $1205 \pm 523$ ms, so trailing leg touchdown constitutes the "early stage." As a result, our PGMs are probabilistic rather than strictly deterministic: the slip types identified could arise from the unconstrained slip itself, reactive strategies at trailing leg touchdown, or both. In contrast, events at slip initiation were exclusively driven by the unconstrained slip.

Unilateral slips (30 total) resulted in 28 recoveries, 2 harness-assisted recoveries, and no falls. By contrast, bilateral slips (28 total) produced 21 recoveries, 1 harness-assisted recovery, and 6 falls, indicating a 21.42% fall rate and higher overall severity than unilateral slips.

Entropy calculations reflected this difference in severity. Bilateral slips (Fig 3) showed a 21.44% greater diversity and "surprise" ($H_{Bilateral} = 6.0992$), while unilateral slips (Fig 4) were less diverse and more predictable ($H_{Unilateral} = 5.0220$) (S2 and S3 Dataset and S1 Fig).

## Slip types

Using both qualitative observations and quantitative foot kinematics after trailing foot touchdown, we identified three slip types that best captured high (Type I), medium (Type II), and low (Type III) severity outcomes (Table 2; Figs 3 and 4). Type I (most severe) involved neither foot being arrested in any plane, allowing both feet to move either in parallel or opposing directions (e.g., both anterior–laterally or one anterior–laterally and the other anterior–contralaterally). This category yielded one harness-assisted recovery in unilateral slips and one harness-assisted recovery plus four falls in bilateral

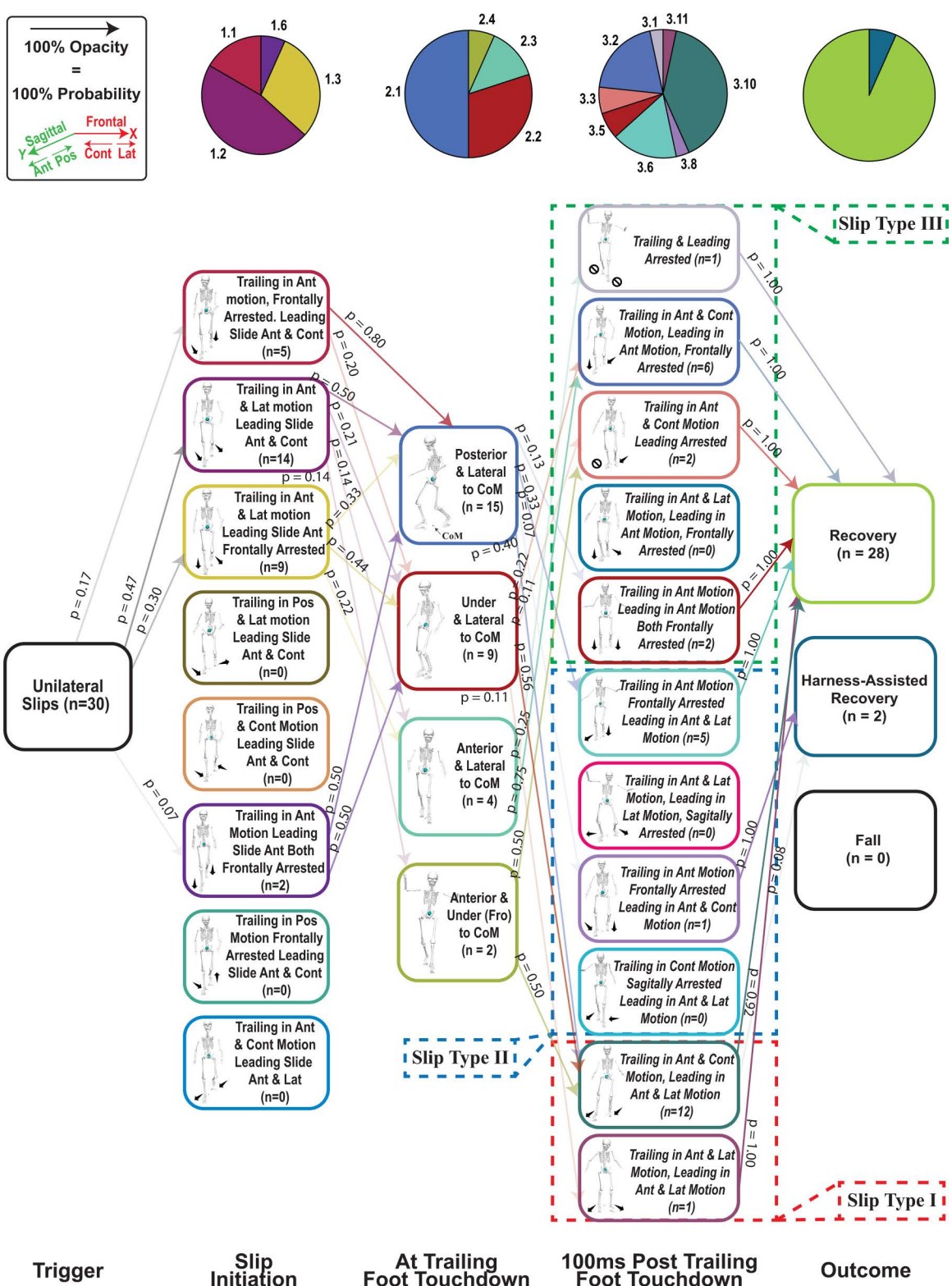

**Fig 3. Probabilistic Graphical Model (PGMs) Describing Unilateral Slips in Both Sagittal and Frontal Planes. p = probability of each event from pre- to post-state. n = number of events for each state.** Pie graphs indicate the percentage of each state ranked in vertical order. Slip types are shown at post trailing leg touchdown based on feet kinematics (Table 2 describes the biomechanical aspects of each slip type).

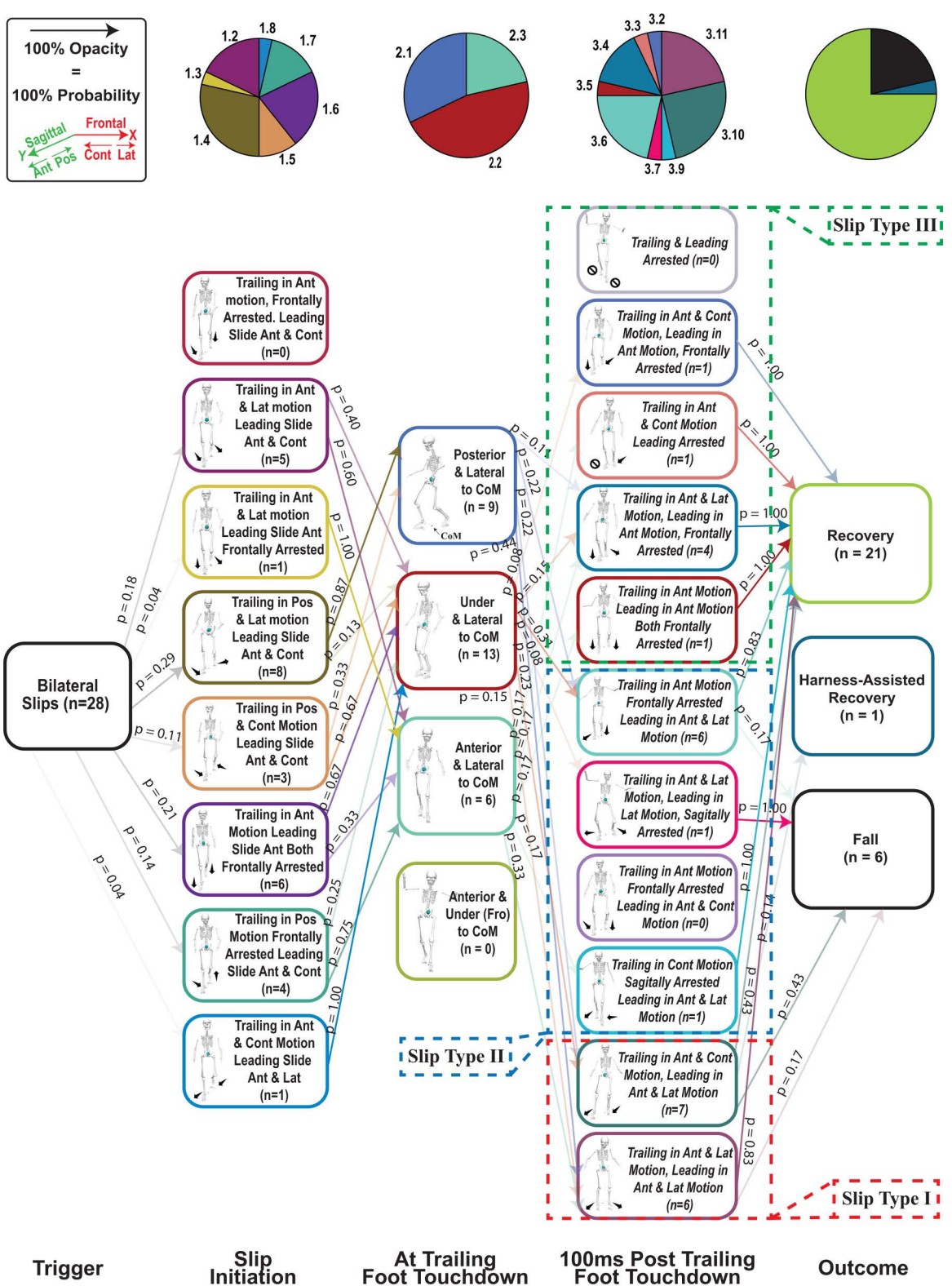

**Fig 4. Probabilistic Graphical Model (PGMs) Describing Bilateral Slips in Both Sagittal and Frontal Planes. p = probability of each event from pre- to post-state. n = number of events for each state.** Pie graphs indicate the percentage of each state ranked in vertical order. Slip types are shown at post trailing leg touchdown based on feet kinematics (Table 2 describes the biomechanical aspects of each slip type).

**Table 2. Slip Types. Refer to Fig 5 for a more visual demonstration.**

| Slip Type | Definition | Severity | Description | 1. Slip Initiation State | 2. Slip State at Trailing Foot Touchdown | 3. Slip State within 100ms Post Trailing Foot Touchdown | Slip Group | Outcome |
|---|---|---|---|---|---|---|---|---|
| Type I | Leading and Trailing Feet Are in frontal and sagittal plane motion | High | Both feet exhibit frontal and sagittal motion after trailing foot touchdown. Feet demonstrate motion on either opposite or similar directions. | 1.1, 1.2, 1.3, 1.4, 1.5, 1.6, 1.7, 1.8 | 2.1, 2.2, 2.3, 2.4 | 3.10) Trailing in anterior and contralateral motion, Leading in anterior and lateral motion (n = 19 [Uni = 12, Bil = 7]) 3.11) Trailing in anterior and lateral motion, Leading in anterior and lateral motion (n = 7 [Uni = 1, Bil = 6]) | Unilateral (n = 13) | Recovery = 12 Harness-Assisted = 1 Fall = 0 |
| | | | | | | | Bilateral (n = 13) | Recovery = 8 Harness-Assisted = 1 Fall = 4 |
| Type II | Leading foot is in frontal plane motion. | Moderate | One foot is arrested in one plane of motion. Leading dominant foot exhibit frontal plane motion after trailing foot touchdown. Trailing non-dominant foot in motion either frontally arrested or in frontal and sagittal motion. | 1.1, 1.2, 1.3, 1.4, 1.5, 1.8 | 2.1, 2.2, 2.3 | 3.6) Trailing in anterior motion frontally arrested, Leading in anterior and lateral motion (n = 11 [Uni = 5, Bil = 6]) 3.7) Trailing in anterior and lateral motion, Leading in lateral motion sagitally arrested (n = 1 [Bil = 1]) 3.8) Trailing in anterior motion frontally arrested, Leading in anterior and contralateral motion (n = 1 [Uni = 1]) 3.9) Trailing in contralateral motion sagitally arrested, Leading in anterior and lateral motion (n = 1 [Bil = 1]) | Unilateral (n = 6) | Recovery = 5 Harness-Assisted = 1 Fall = 0 |
| | | | | | | | Bilateral (n = 8) | Recovery = 6 Harness-Assisted = 0 Fall = 2 |
| Type III | Leading foot is not in frontal plane motion | Low | Leading dominant foot is either frontally or completely arrested. | 1.2, 1.3, 1.4, 1.7 | 2.1, 2.2, 2.3, 2.4 | 3.1) Trailing Arrested, Leading Arrested (n = 1 [Uni = 1]) 3.2) Trailing in anterior and contralateral motion, Leading in anterior motion frontally arrested (n = 7 [Uni = 6, Bil = 1]) 3.3) Trailing in anterior and contralateral motion, Leading arrested (n = 3 [Uni = 2, Bil = 1]) 3.4) Trailing in anterior and lateral motion, Leading in anterior motion frontally arrested (n = 4 [Bil = 4]) 3.5) Trailing in anterior motion frontally arrested, Leading in anterior motion frontally arrested (n = 3 [Uni = 2, Bil = 1]) | Unilateral (n = 11) | Recovery = 11 Harness-Assisted = 0 Fall = 0 |
| | | | | | | | Bilateral (n = 7) | Recovery = 7 Harness-Assisted = 0 Fall = 0 |

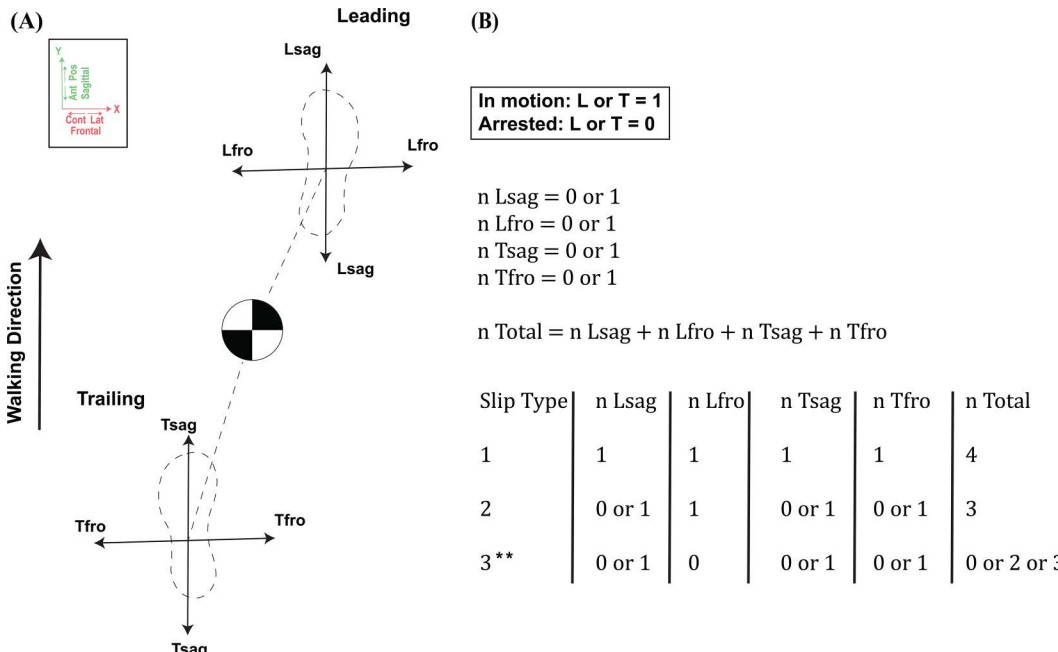

**Fig 5. Slip Types Demonstration & Calculations.** (A), Stick figure demonstration of the two planes of motion that the Leading (L) and Trailing (T) feet can travel in. Four planes of motion emerge in total: sagittal and frontal planes for each foot. The black and white circle indicates the CoM. (B), Method used to calculate and categorize the three slip types. When the foot's velocity was lower than 0.099 m/s, then the foot was considered arrested and was assigned n = 0, and the foot was considered in motion when its velocity was greater than 0.01 m/s and was assigned n = 1. N total was used to differentiate between the slip types. Slip type I includes all trials when both feet where in motion in all four planes after trailing leg touchdown. Slip type II include all trials when the dominant foot was in frontal plane motion, and three planes are in motion simultaneously. Slip type III include all trials when the dominant foot was arrested in frontal plane, where the foot can either be completely arrested or in sagittal plane motion only. **Although slip type III includes trials with three planes simultaneously in motion, unlike slip type II, the dominant leading foot was always frontally arrested.

slips. Type II (moderately severe) occurred when only one foot was arrested in one plane: after trailing foot touchdown, the leading (dominant) foot continued moving in the frontal plane, while the trailing (non-dominant) foot was either frontally arrested or moved in both frontal and sagittal directions. Consequently, Type II produced one harness-assisted recovery in unilateral slips and two falls in bilateral slips. In Type III (least severe), the dominant foot was either fully arrested or confined to frontal-plane motion, leading to complete recoveries for all trials (11/11 unilateral; 7/7 bilateral). Although more detailed slip types could be defined by incorporating additional kinematic parameters, our primary goal was to use PGMs to categorize outcomes (fall, harness-assisted, or recovery) based on each foot's position and motion following trailing foot touchdown (Fig 5).

### Differences in L across slip types

Disaggregating slipping trials by slip type and comparisons between slip types showed significant differences in L. Multivariate GLMs showed a significantly different transverse L at slip initiation ($F_{(5)} = 2.409$, $p = .049$), frontal ($F_{(5)} = 2.530$, $p = .040$) and sagittal ($F_{(5)} = 2.399$, $p = .049$) L at trailing foot touchdown, and sagittal L within 100ms post trailing foot touchdown ($F_{(5)} = 8.628$, $p < .001$) across slip types and slip groups (Figs 6 and 7 – S4 Dataset). Specifically, pairwise comparison within unilateral slips showed significantly different sagittal L at trailing foot touchdown between slip type I and III ($p = 0.038$, 95% CI [0.339, 11.290]), and significantly different frontal ($p = 0.040$, 95% CI

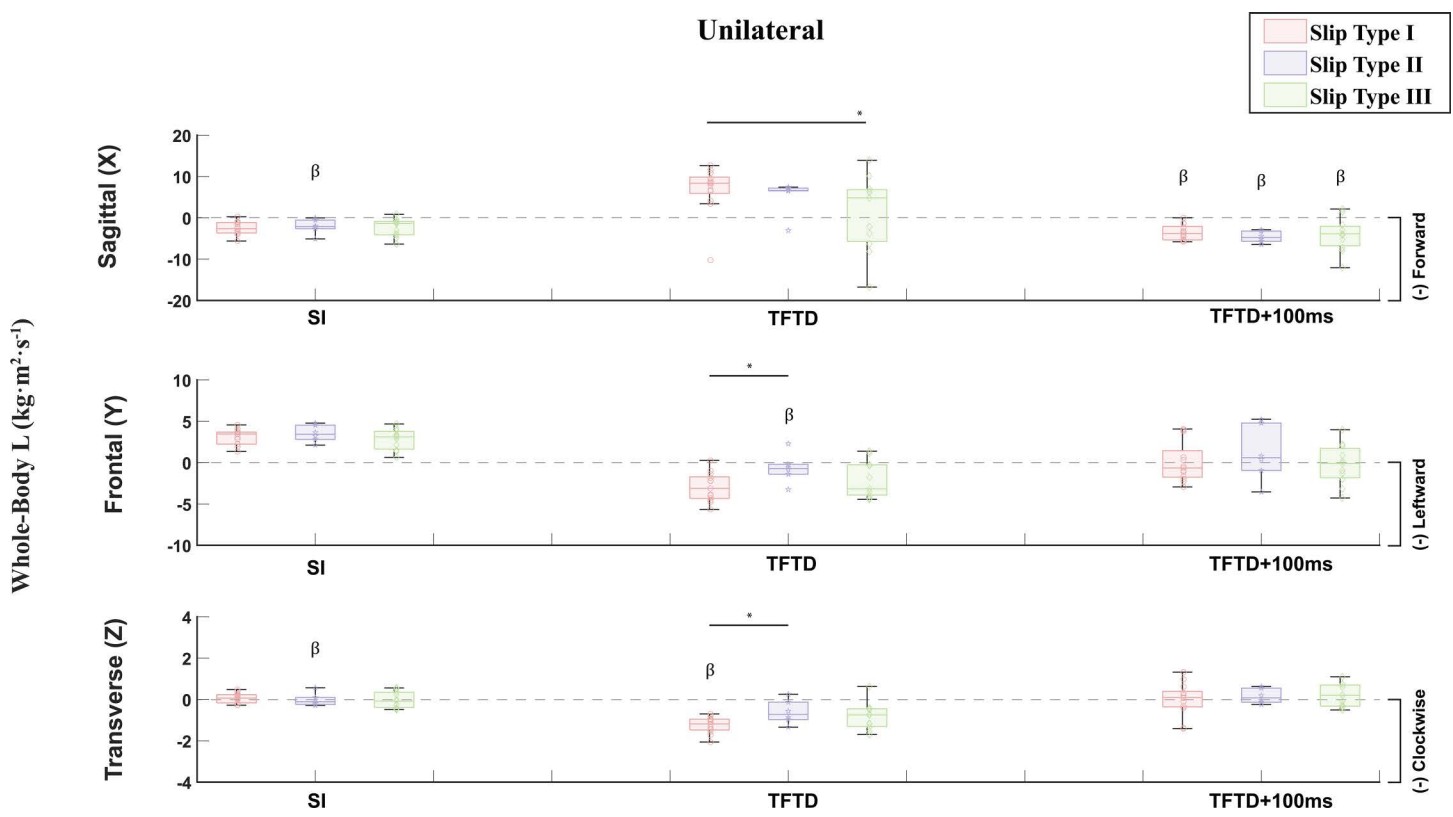

**Fig 6. L for the Unilateral slip group at slip initiation (SI), trailing foot touchdown (TFTD) and 100ms post trailing foot touchdown (TFTD+100ms) compared by the three defined slip types and presented in the sagittal, frontal, and transverse planes.** Angular Momenta are displayed in the form of histograms providing the median, the lower and upper quartiles, the minimum and maximum values, and outliers if any are available. β: Statistically Significant Differences between Unilateral and Bilateral Slips. *p < 0.05.

[−4.484, −0.105]) and transverse (p = 0.031, 95% CI [−1.220, −0.059]) L at trailing foot touchdown between slip types I and II (Fig 6 - S4 Dataset). Pairwise comparisons within bilateral slips at slip initiation showed significantly different sagittal L between slip types II and III (p = 0.019, 95% CI [−5.262, −0.496]), significantly different frontal L between slip types I and III (p = 0.011, 95% CI [0.398, 2.888]) and between slip types II and III (p = 0.004, 95% CI [0.712, 3.461]), and significantly different transverse L (p = 0.023, 95% CI [0.078, 1.022]) between slip types II and III (Fig 7 - S4 Dataset). At trailing foot touchdown, pairwise comparisons within bilateral slips showed significantly different frontal L between slip types I and III (p = 0.016, 95% CI [−4.663, −0.505]), and between slip types II and III (p = 0.020, 95% CI [−5.041, −0.450]) (Fig 7 – S4 Dataset). Moreover, pairwise comparisons between slip groups across slip types revealed statistically significant differences between unilateral and bilateral slip groups in sagittal (Slip Type II: p = 0.038, 95% CI [0.147, 5.121]) and transverse (Slip Type II: p = 0.039, 95% CI [−1.014, −0.028]) L at slip initiation, frontal (Slip Type II: p = 0.026, 95% CI [0.346, 5.137]) and transverse (Slip Type I: p = 0.003, 95% CI [−1.178, −0.256]) L at trailing foot touchdown, and sagittal L (Slip Type I: p < 0.001, 95% CI [−10.268, −4.212], Slip Type II: p = 0.002, 95% CI [−10.815, −2.477], Slip Type III: p = 0.029, 95% CI [−7.951, −0.485]) 100ms post trailing foot touchdown (Figs 6 and 7 – S4 Dataset). According to Cohen's $\eta^2$, all GLMs showed very strong effect sizes (Table 3).

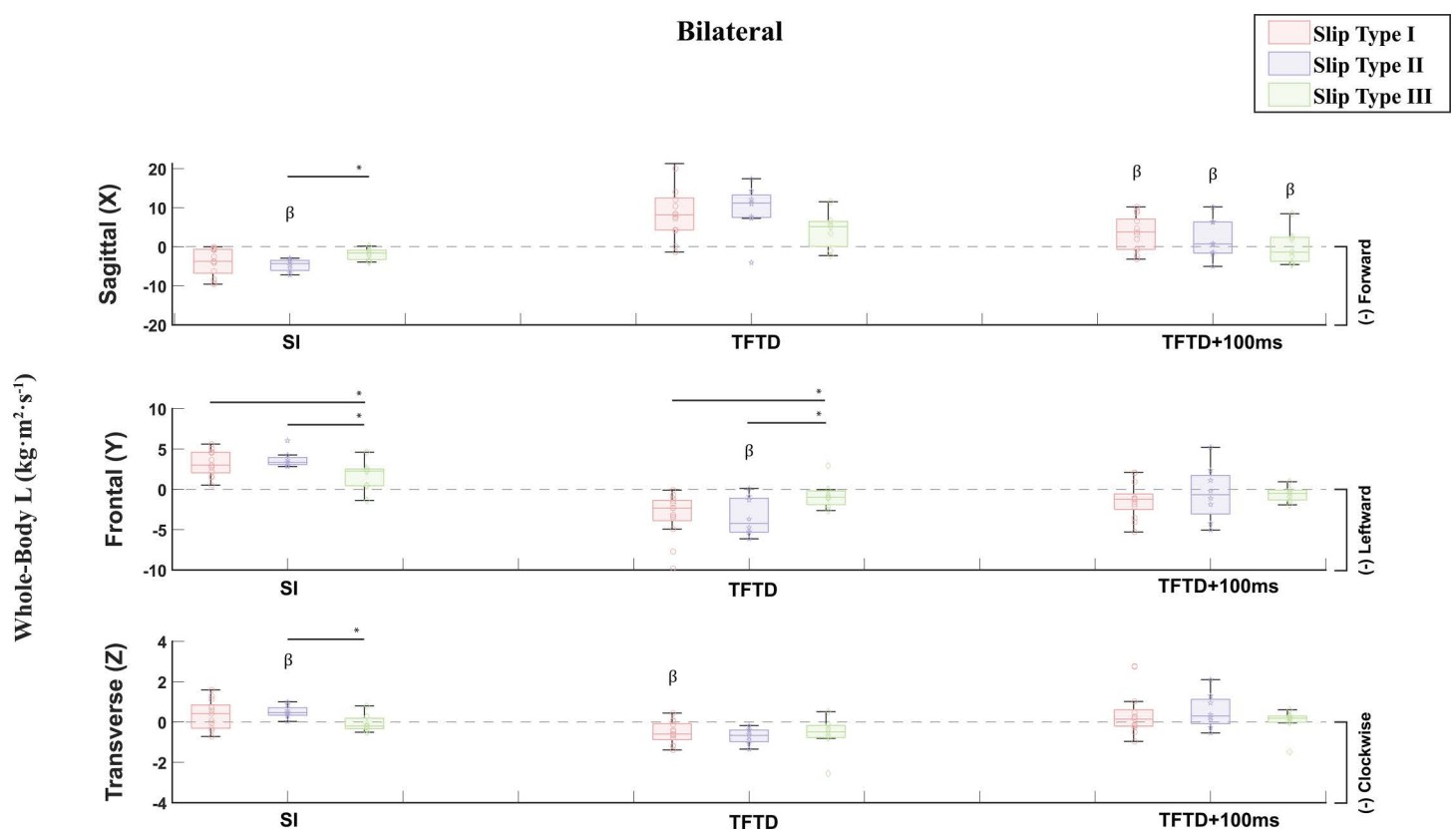

**Fig 7. L for the Bilateral slip group at slip initiation (SI), trailing foot touchdown (TFTD) and 100ms post trailing foot touchdown (TFTD+100ms) compared by the three defined slip types and presented in the sagittal, frontal, and transverse planes.** Angular Momenta are displayed in the form of histograms providing the median, the lower and upper quartiles, the minimum and maximum values, and outliers if any are available. β = Statistically Significant Differences between Unilateral and Bilateral Slips. *p < 0.05.

## Differences in feet velocities across slip types

Disaggregating slipping trials by slip type and comparisons between slip types showed significant differences in average feet velocities from trailing foot touchdown until 100ms post trailing foot touchdown. Multivariate GLMs showed significantly different leading foot velocities in the frontal (F(5) = 9.539, p < .001) and sagittal (F(5) = 3.972, p = .004) planes, and significantly different trailing foot velocity in the sagittal plane (F(5) = 5.507, p < .001) across slip types and slip groups (Fig 8 – S5 Dataset). Specifically, pairwise comparison within unilateral slips showed significantly different leading foot velocities in the frontal plane between slip type I and III (p < 0.001, 95% CI [0.138, 0.412]), and slip II and III (p < 0.001, 95% CI [0.119, 0.458]), as was expected since the dominant foot was frontally arrested in slip type III (Fig 8 – S5 Dataset). Within the unilateral slip group, the dominant foot velocity in the sagittal plane also showed significant differences between slip types I and III (p = 0.023, 95% CI [0.078, 0.989]) (Fig 8 – S5 Dataset). Pairwise comparisons within bilateral slips also showed significantly different leading foot velocities in the frontal plane between slip type I and III (p < 0.001, 95% CI [0.184, 0.497]), and slip II and III (p < 0.001, 95% CI [0.199, 0.544]), as was expected since the dominant foot was frontally arrested in slip type III (Fig 8 – S5 Dataset). Within the bilateral slip group, leading foot velocity in the sagittal plane also showed significant differences between slip types I and III (p = 0.002, 95% CI [0.319, 1.360]), and trailing foot velocity in the sagittal plane showed significant differences between slip types I and II (p = 0.028, 95% CI [0.092, 1.567]) (Fig 8 – S5 Dataset). Moreover, pairwise comparisons between slip groups across slip types revealed statistically significant

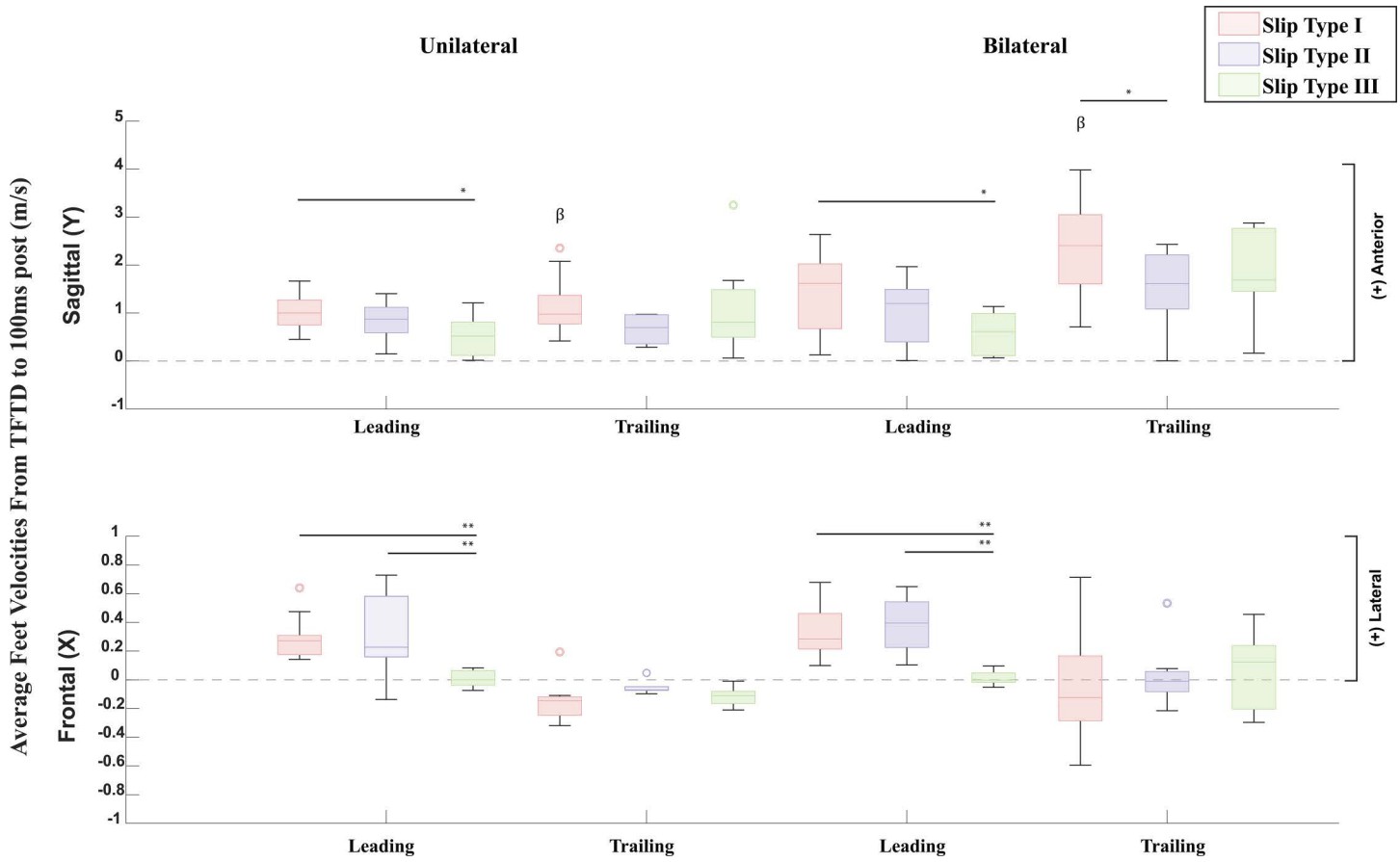

**Fig 8. Average Leading and Trailing Feet Velocities from trailing foot touchdown until 100ms post trailing foot touchdown compared by the three defined slip types and two slip groups and presented in the sagittal and frontal planes.** Feet velocities are displayed in the form of histograms providing the median, the lower and upper quartiles, the minimum and maximum values, and outliers if any are available. β = Statistically Significant Differences between Unilateral and Bilateral Slips. *p < 0.05 **p < 0.001.

**Table 3. Effect Sizes ($\eta^2$) for the significant GLMs.** $\eta^2 = 0.01$ is a weak effect, $\eta^2 = 0.059$ is a moderate effect, $\eta^2 = 0.138$ is a strong effect, and anything above is a very strong effect. Observed power column indicates the statistical power for each GLM.

| GLMs | | Observed Power | Effect Size | |
|---|---|---|---|---|
| GLM # | Dependent Variable | Partial $\eta^2$ | Interpretation | |
| First GLM | Transverse L at SI | 0.719 | 0.188 | Very Strong |
| | Sagittal L at TLTD | 0.717 | 0.187 | Very Strong |
| | Frontal L at TLTD | 0.743 | 0.196 | Very Strong |
| | Sagittal L 100ms post TLTD | 1.000 | 0.453 | Very Strong |
| Second GLM | Leading Foot Sagittal Velocity | 0.925 | 0.276 | Very Strong |
| | Leading Foot Frontal Velocity | 1.000 | 0.478 | Very Strong |
| | Trailing Foot Sagittal Velocity | 0.984 | 0.346 | Very Strong |

differences between bilateral and unilateral slip groups in trailing foot sagittal velocity within slip type I (p < 0.001, 95% CI [0.574, 1.862]) (Fig 8 – S5 Dataset). According to Cohen's $\eta^2$, all GLMs showed very strong effect sizes (Table 3).

## Discussion

In this study, we tested whether unconstrained bilateral slips would (1) be more diverse and (2) pose a higher fall risk than unilateral slips. By reducing friction under both feet, the leading and trailing feet could slip in both anteroposterior and mediolateral directions, increasing the potential variety of slip paths. We used a probabilistic graphical model to capture these stochastic patterns and compute an entropy measure, which confirmed higher entropy values for bilateral slips, indicating greater diversity and uncertainty about how a bilateral slip might unfold.

Regarding fall risk, the bilateral slip group experienced six falls (21.42%), whereas the unilateral slip group exhibited none. This outcome aligns with our second hypothesis, suggesting bilateral slips are more likely to result in falls under these experimental conditions. The absence of falls in the unilateral slip group further implies that a single-foot slip may not be sufficiently challenging to induce falls in a younger adult population and thus may not capture the full scope of fall risk in laboratory studies.

Robotics-based balance control models can provide a useful framework for understanding why bilateral slips are more hazardous and variable than unilateral slips. In these models, falls occur when a disturbance pushes the body's center of mass (CoM) outside its "recoverability region," defined as the range of positions and velocities from which balance can still be restored without an extra corrective step [65,66]. During a unilateral slip, the non-slipping foot maintains stable contact with the ground, enabling rapid generation of corrective ground forces to steer the CoM back into this recoverability region. In contrast, a bilateral slip removes traction from both feet simultaneously, leaving no stable limb available to produce corrective forces. Consequently, the CoM moves farther, exits the recoverability region more frequently, and results in both higher fall rates and greater variability in outcomes. Critically, robotic platforms serve as physical models for rigorously testing causal hypotheses about balance control strategies, experiments that would be impractical or unsafe with human subjects. By systematically varying control strategies on these robotic or exoskeleton-based models, researchers can identify precisely which torque or stepping interventions effectively reduce slip distances and restore the CoM within recoverable limits. Recent exoskeleton studies, for example, show that targeted knee joint torques substantially reduce slip distances and prevent falls [67], directly validating recoverability-region concepts and highlighting why bilateral slips represent a uniquely severe challenge to human balance.

Our third hypothesis posited that distinguishing slips into different qualitative categories of foot-movement directions would reveal differences in slip severity, as indicated by L and slipping velocities. Accordingly, both unilateral and bilateral slips were divided into three slip types, each characterized by a distinct total number of planes of movement for the leading and trailing feet. These qualitatively defined types displayed quantitative differences in L at slip initiation, trailing foot touchdown, and 100 ms post-touchdown (Figs 6 and 7; S4 Dataset). Moreover, an additional analysis of foot velocities (Fig 8; S5 Dataset) confirmed statistically significant differences among the three slip types: Type I proved most severe, Type III was least severe, and Type II was intermediate. Notably, in Type I, the bilateral slip group exhibited higher trailing-foot velocities than the unilateral group, suggesting that bilateral slips may be more hazardous under these high-severity conditions.

Many previous studies have categorized slips by type, yet we contend that measuring slip diversity ideally requires unconstrained slips, where the foot can move in both anteroposterior and mediolateral directions. Such unconstrained slips arise naturally in environments with friction lower than the minimum needed for stable walking (μ = 0.2) [68]. In contrast, constrained slips, typically limited to the sagittal plane, are likely to produce findings that reflect the imposed laboratory conditions rather than the natural mechanics of a foot sliding across a low-friction surface. While examining constrained and unconstrained slips in laboratory settings remains valuable for understanding recovery strategies, our data suggests that many real-world slip-induced injuries and fatalities [2] stem from unconstrained bilateral slips. Consequently,

training balance recovery using only constrained slips or strictly unilateral slips may not fully prepare individuals to respond to the more hazardous, real-life scenarios in which both feet can slip simultaneously.

Should we intentionally limit the slipping feet's degrees of freedom by "arresting" motion in one or more directions for unconstrained slips? In this study, we classified slip types by the total number of directions, anteroposterior and medio-lateral, in which the leading and trailing feet were moving. For instance, if both feet traveled in both planes, totaling four directions, the slip was classified as Type I. Notably, most falls (4 of 6) occurred in Type I, whereas fewer (2 of 6) occurred in Type II, where at least one direction was always arrested (e.g., restricting the leading foot's frontal-plane movement). This pattern suggests that arresting foot motion in at least one direction may improve recovery, and further restricting additional directions could enhance it even more. However, future experiments that deliberately manipulate the number of degrees of freedom available to each foot are needed to establish a causal link between these slip types and the likeli-hood of falls.

In this experiment, unilateral unconstrained slips did not induce falls in our sample of healthy younger adults. Although our probabilistic graphical model approach cannot fully explain the absence of falls here, we can compare our biome-chanical data with that reported for sagittally constrained slips [14–27], as well as for unconstrained slips [4,9,31–47]. Under unconstrained conditions, researchers often lubricate the floor so that participants slip forward on the leading leg, sometimes placing the trailing leg on the same low-friction surface and thus creating a bilateral slip. Bilateral slips have also been induced using two sliding platforms, one for the leading foot at heel strike and another for the trailing foot at touchdown. These data allow direct comparisons of our unconstrained bilateral slips with those previously classified as constrained bilateral slips.

Our findings, showing that in less severe slips, the leading and/or trailing foot often slows or becomes fully arrested after trailing leg touchdown (Fig 8; S5 Dataset), and that bilateral slips increase slip diversity and severity, align with both unconstrained [35,46] and sagittally constrained slip studies [27]. In unconstrained slips, older adults who placed their recovery foot on a slippery surface more frequently incurred bilateral slips that increased posterior displacement of the CoM and higher slip velocities, leading to more frequent falls [46]. Recent work on unconstrained unilateral slips also found that although 95% of recovery trials involved trailing-leg contact with a contaminated floor (to avoid split-feet falls), nearly all feet-forward and lateral falls similarly placed the trailing foot on the contaminated floor, thus creating bilateral slips [31]. Higher trailing-foot velocities in both the sagittal and frontal planes may raise the likelihood of a bilat-eral slip-induced fall compared to lower-velocity recoveries [31]. Sagittally constrained slip-perturbation studies reinforce the importance of arresting the feet after trailing foot touchdown to lower slip severity and fall risk [14–29,69]. Notably, sagittally constrained bilateral slips yield higher fall rates than their unilateral counterparts [27], likely because platform perturbations drive both feet forward, increasing slip severity relative to unilateral slips, which commonly result in split-feet falls. When participants in these studies inadvertently missed the second platform, foot velocities and fall rates were lower despite both feet translating forward [27], underscoring the added risk of bilateral slips. In line with this, more severe unilateral unconstrained slips exhibit higher sagittal L [48]. Although no prior studies to our knowledge have compared L between unilateral and bilateral slips, our data show significantly higher sagittal, frontal, and transverse L in bilateral slips compared to unilateral slips. Moreover, slip Types I and II manifested higher frontal L at slip initiation than Type III, as well as higher frontal and sagittal L than Type III overall, and Type I further showed increased sagittal L 100 ms after trailing leg touchdown (Figs 6–7; S4 Dataset). These patterns collectively indicate greater slip severity for bilateral versus unilateral slips, and for Types I and II compared to Type III.

Previous work on unconstrained unilateral slips suggests that lateral recovery-foot placement relative to the CoM can better predict falls and recoveries than foot velocities alone [46]. In contrast, sagittally constrained unilateral slips show an increased likelihood of recovery when the recovery foot lands within 30% of foot length behind the CoM [28]. Although our findings also emphasize the importance of trailing leg touchdown position in redirecting the backward angular impulse, our data indicate that, for the unconstrained slips we studied, post–trailing leg touchdown foot velocities carried more weight

in determining slip severity than touchdown position per se. This aligns with research on constrained slips, which reports that while early slip-phase kinematics (heel strike to trailing foot swing initiation) are similar across different outcomes, later slip-phase kinematics (trailing foot swing initiation to ~110 ms post-touchdown) diverge among recovery, split-fall, and feet-forward fall trials [69]. Factors such as larger trunk flexion, leading leg knee flexion and plantarflexion, and trailing leg knee extension have all been associated with reduced fall incidence [69]. Here, however, we used probabilistic rather than deterministic models to classify slip types, relying on foot behavior and outcome severity under naturally unconstrained conditions. We also focused on a short time window (100 ms post–trailing leg touchdown) rather than the entire slip time-line, so future studies incorporating full-body kinematic and kinetic analyses from slip initiation to slip termination could more directly pinpoint deterministic factors that distinguish falls from recoveries.

Using unconstrained slips is vital for capturing a broad range of slip states and categorizing them into distinct types; however, our study has several limitations. First, the small sample size, restricted to younger adults, constrains generalizability and may have precluded identification of additional slip states. A higher-risk population (e.g., older adults, amputees, individuals with Parkinson's disease, or those who have undergone joint replacements) might exhibit slip behaviors beyond those observed here. Second, because we did not directly control slip types but merely observed them as outcomes of unconstrained slips, we cannot infer strict causal relationships between slip types and falls. Constrained slips can target specific behaviors (e.g., feet-forward or split falls), whereas unconstrained slips on low-friction surfaces reflect more natural, multifaceted slip mechanics. Our goal, however, was not to induce particular behaviors but to investigate how unconstrained slips unfold under real-world–like conditions. Third, to avoid overfitting a complex model to a small dataset, we focused on the early slip timeline (up to 100 ms after trailing leg touchdown), leaving out kinematics that develop once the participant begins reacting to the slip. Similarly, any trials without frontal-plane foot motion were grouped into a single slip type (Type III), even though more detailed categories (e.g., backward loss of balance, skate-over recovery) have been described in the literature. Finally, although our definition of slip initiation (occurring <30 ms after WASP trigger) ensures that any foot movement in that window can be attributed to the slip itself, the 100 ms post–touchdown window might reflect both the WASP trigger and early reactive responses. Given these limitations, future research could explore slip detection and balance-recovery techniques inspired by recent robotics findings. For instance, rapid disturbance detection methods developed for robotics have demonstrated potential for quickly restoring balance on very low friction surfaces [70]. Moreover, wearable sensors derived from robotic slip-detection research have achieved high real-time accuracy in objectively classifying slips [71–73]. Investigating new approaches in human slip scenarios could provide further insights into reducing the severity and high fall incidence associated with bilateral slips observed here.

## Conclusions

Our results demonstrate that sudden, unconstrained bilateral slips are both more diverse and riskier than their unilateral counterparts. In line with our initial hypotheses, bilateral slips yielded a higher fall rate (21.42%) and displayed more than twice as many possible slip events (unilateral: 32 vs. bilateral: 68). We also observed that partially arresting either the leading or trailing foot in at least one plane of motion reduces fall risk; notably, arresting the leading foot in the frontal plane eliminated falls in both unilateral and bilateral slips. To our knowledge, this study is the first to classify unconstrained unilateral and bilateral slips into slip groups using PGMs that integrate qualitative and quantitative biomechanical data. Future investigations should explore whether different slip contexts (unilateral vs. bilateral) and slip types require distinct motor coordination strategies for successful recovery.

## Supporting information

**S1 Dataset.** Participants' History of Falls.
(XLSX)

**S1 Table. Biomechanical description of the Unilateral and Bilateral Slips' Probabilistic Graphical Models (PGMs).**
(PDF)

**S2 Dataset. Slip Entropy Calculation.**
(PDF)

**S3 Dataset. Unilateral and Bilateral Slip Entropy Data.**
(XLSX)

**S1 Fig. Differences in the Entropy and the Possibility of a Single Event between Unilateral and Bilateral Slips.**
(PDF)

**S4 Dataset. Differences in L across slips types.**
(XLSX)

**S5 Dataset. Differences in feet velocities across slips types.**
(XLSX)

## Acknowledgments

The authors would like to thank Drs. Jaap van Dieën, Mark Grabiner, Alfred Fisher, and Mukul Mukherjee for their thoughtful feedback and manuscript revisions.

## Author contributions

**Conceptualization:** Abderrahman Ouattas, Corbin M. Rasmussen, Nathaniel H. Hunt.

**Data curation:** Abderrahman Ouattas, Andrew Walski, Seongwoo Mun.

**Formal analysis:** Abderrahman Ouattas, Nathaniel H. Hunt.

**Funding acquisition:** Abderrahman Ouattas, Corbin M. Rasmussen, Nikolaos Stergiou, Nathaniel H. Hunt.

**Investigation:** Abderrahman Ouattas, Andrew Walski, Seongwoo Mun, Corbin M. Rasmussen, Nikolaos Stergiou, Nathaniel H. Hunt.

**Methodology:** Abderrahman Ouattas, Seongwoo Mun, Corbin M. Rasmussen, Nikolaos Stergiou, Nathaniel H. Hunt.

**Project administration:** Abderrahman Ouattas, Seongwoo Mun, Nathaniel H. Hunt.

**Resources:** Abderrahman Ouattas, Andrew Walski, Nikolaos Stergiou, Nathaniel H. Hunt.

**Software:** Abderrahman Ouattas, Nathaniel H. Hunt.

**Supervision:** Abderrahman Ouattas, Nikolaos Stergiou, Nathaniel H. Hunt.

**Validation:** Abderrahman Ouattas, Nathaniel H. Hunt.

**Visualization:** Abderrahman Ouattas, Seongwoo Mun, Corbin M. Rasmussen, Nathaniel H. Hunt.

**Writing – original draft:** Abderrahman Ouattas, Nathaniel H. Hunt.

**Writing – review & editing:** Abderrahman Ouattas, Andrew Walski, Seongwoo Mun, Corbin M. Rasmussen, Nikolaos Stergiou, Nathaniel H. Hunt.

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
