## [Decision Letter · Decision Letter 0]

30 Apr 2025

PONE-D-25-09847Higher Fall Rates and Broader Kinematic Diversity in Bilateral Versus Unilateral Unconstrained SlipsPLOS ONE

Dear Dr. Ouattas,

Thank you for submitting your manuscript to PLOS ONE. After careful consideration, we feel that it has merit but does not fully meet PLOS ONE’s publication criteria as it currently stands. Therefore, we invite you to submit a revised version of the manuscript that addresses the points raised during the review process.

While the reviewers had overall positive reviews of the paper, there are several areas that need revision.

We look forward to receiving your revised manuscript.

Kind regards,

Anne E. Martin

Academic Editor

PLOS ONE

Journal Requirements:

2. Thank you for stating the following financial disclosure: [This work was supported by the National Institutes of Health; NIH 2P20 GM109090-06, NIH R15AG063106-01, and NIH NIGMS P20GM152301.]. 

3. Thank you for stating the following in the Acknowledgments Section of your manuscript: [The authors would like to thank Drs. Jaap van Dieën, Mark Grabiner, Alfred Fisher, and Mukul Mukherjee for their thoughtful feedback and manuscript revisions. This work was supported by the National Institutes of Health; NIH 2P20 GM109090-06, NIH R15AG063106-01, and NIH NIGMS P20GM152301.]

Please remove any funding-related text from the manuscript and let us know how you would like to update your Funding Statement. Currently, your Funding Statement reads as follows: [This work was supported by the National Institutes of Health; NIH 2P20 GM109090-06, NIH R15AG063106-01, and NIH NIGMS P20GM152301.]. 

4. In the online submission form, you indicated that [The raw data supporting the conclusions of this article will be made available by the authors, without undue reservation.].

5. We note that there is identifying data in the Supporting Information file <SupplementaryDataSheet_1_ParticipantsDemographicsHistoryofFalls.xlsx>. Due to the inclusion of these potentially identifying data, we have removed this file from your file inventory. Prior to sharing human research participant data, authors should consult with an ethics committee to ensure data are shared in accordance with participant consent and all applicable local laws.

-Location data

Please remove or anonymize all personal information (ID,AGE,)ensure that the data shared are in accordance with participant consent, and re-upload a fully anonymized data set. Please note that spreadsheet columns with personal information must be removed and not hidden as all hidden columns will appear in the published file.

6. Please include captions for your Supporting Information files at the end of your manuscript, and update any in-text citations to match accordingly. Please see our Supporting Information guidelines for more information: http://journals.plos.org/plosone/s/supporting-information .

Reviewers' comments:

Reviewer's Responses to Questions

**Comments to the Author**

1. Is the manuscript technically sound, and do the data support the conclusions?

Reviewer #1: Yes

Reviewer #2: Yes

2. Has the statistical analysis been performed appropriately and rigorously? 

Reviewer #1: Yes

Reviewer #2: Yes

3. Have the authors made all data underlying the findings in their manuscript fully available?

Reviewer #1: Yes

Reviewer #2: Yes

4. Is the manuscript presented in an intelligible fashion and written in standard English?

Reviewer #1: Yes

Reviewer #2: Yes

5. Review Comments to the Author

Reviewer #1: The manuscript presents interesting results that are of interest to biomechanics and wider community as it relates to slip-induced falls. Specifically, the study presents findings that the bilateral slips are more complex in nature and result in higher fall rates compared to the unilateral slips.

General comments: While the study is really interesting, as a reader/reviewer, I had a very hard time comprehending all the information, due to repeated information/sentences and multivariable analysis with many results presented all together. I suggest to reorganize the results by including subsections and emphasizing the most important results within each of them. In addition, selection of specific timings or events was not well justified. The whole analysis and results comparison was based on those, which requires clarification or standardize representation (and/or potentially rerunning the analysis).

Below are provided some additional comments that the authors should address to improve the clarity.

1. The last two sentences of the first paragraph in Intro are slightly ambiguous. Are you making comparison of current numbers to those or are you referring to words in their papers. Please rephrase.

2. Please try to avoid repeating information. for example, shorten last sentence before Section 4 and also spell out angular momentum instead of using L. Note that you defined L as a specific variable. Similarly, see paragraph 2 on page 13 for repeating sentences or definitions

3. P.7 Line158: Ideally, you should reorganize the Subfigures in Fig. 1 to call this Fig. 1A as it appears first. Simply switch A and B subfigures.

4. Why were slips induced during 0-33% of stance phase and not at the heel strike? This does not seem the same as most commonly reported in the literature. Please provide supporting statement/reference from existing literature.

5. P8 Line 176: Were during bilateral slips both WASPs activated simultaneously? It seems not, if leading foot was during early and trailing foot was during late stance slip? Please clarify and revise that paragraph. Why was WASP not actuated at the foot touch down? Additionally, how does the instructions of continuing to walk vs stop walking after slip onset effect the end results?

6. P9. Did the authors check that the gait pattern returned to a normal walking gait after the slip? How did the authors guarantee that the subjects did not alter the gait as to expect more slip perturbations?

7. Why was slipping timeline selected as the slip onset. Why it was not measured from the heel strike to have an objective measure and for comparison between all slips that may start at different gait instances? I would suggest to revise that or provide justification.

8. Why focusing only within 30 msec of slip initiation or 30msec of trailing limb touchdown? Why is H-reflex latency important for determining foot velocities? Does this has to do anything with friction demands and eliminates effects of human reflexes?

9. P11 Line 228: It is unclear what the authors are trying to imply here by 'because whole-body angular impulse..' Please explain how is angular impulse used and calculated in this study as a metric.

10. Please check the text for any typos or misspelling. For example, P11, Line 245 should be trailing foot.

11. Please provide justification how was chosen 100ms after trailing limb touchdown as a relevant.

12. Please check if m_n is really a vector or it is a scalar in Equation 2.

13. P13&14. If equations are not numbered, please insert them in text directly.

14. P14 Line 300: Please do not repeat all details every time (Simply say..in all planes, X, Y, and Z OR in sagittal, frontal and transverse plane) it is enough.

15. Define abbreviation only once. See PGM.

16. The discussion is broad and well justified.

Reviewer #2: The paper presents experimental studies on unexpected, unstrained bilateral slip during walking gaits to answer three questions such as probability of bilateral and unilateral slip induced falls, diversity of bilateral and unilateral slip-and-fall, and severity between bilateral and unilateral slips. The experiments were well thought and developed and the results are sound. The paper is clearly written and presented and this reviewer enjoyed reading the manuscript. I have several comments and suggestions.

1. The authors used wearable apparatus for slipping perturbations (WASP) to create foot slip in experiments. I felt that the paper lacks detailed discussion of WASP. Maybe the authors published WASP previously and it would still be helpful to briefly discuss and mention this to keep the paper self-contained. Particularly, it is unclear whether the slip-and-fall gait by WASP would represent the same biomechanics and natural human gait and responses under unexpected foot slip.

2. Most existing slip related literature focused on sagittal slip conditions. It would be helpful to point out and clarify how much and how often the lateral slip (so called unconstrained slip in this work) actually happen in real life.

3. The authors used probabilistic graphical models (PGMs) and general linear models (GLMs) to conduct statistical analyses. The authors should clarify why these models are used particularly for this work. The benefits and rationales should be discussed compared with other methods.

4. Although the paper reviewed most relevant work on slip-and-fall studies, it seems that the authors did not include some studies using dynamics model and analysis (e.g., engineering and robotics). For example, the work by Trkov et al. (2018), "Shoe-floor interactions in human walking with slips: Modeling and experiments." ASME Journal of Biomechanical Engineering, 140: article 031005; Mihalec et al. (2022), "Balance recoverability and control of bipedal walkers with foot slip." ASME Journal of Biomechanical Engineering, 144(5): article 051012; Zhu and Yi (2023), "Knee exoskeleton-enabled balance control of human walking gait with unexpected foot slip." IEEE Robotics and Automation Letters, 8(11): 7751-7758, to list a few. The authors might conduct more extensive literature review to include more recent studies in engineering field for the related work.

5. The authors might clarify and justify the some used parameters in experimental protocols and analysis. For example,

a) It seems that the paper still uses peak slip velocity as metric to define "slip severity". However, this might not be comprehensive and unconclusive in literature. The authors took whole-body angular momentum as one additional metric to evaluate the slip response and that is beneficial. It would be helpful for the authors to make a clear statement on how to define severity in this study in Introduction section.

b) It is unclear to this reviewer on how several critical parameters (e.g., 100 ms trailing leg touchdown, etc.) was chosen for analysis and calculation. It is helpful for the authors to further clarify these selections.

6. PLOS authors have the option to publish the peer review history of their article (what does this mean? ). If published, this will include your full peer review and any attached files.

**Do you want your identity to be public for this peer review?** For information about this choice, including consent withdrawal, please see our Privacy Policy .

Reviewer #1: No

Reviewer #2: **Yes: ** Jingang Yi

---

## [Author Response · Author response to Decision Letter 1]

13 Jun 2025

General comments to the editors:

We would like to thank all PLOS ONE editors, members, and staff for bringing these points to our attention. The following is our response to each requirement.

• Thank you for your feedback and revisions. We revised the manuscript to meet PLOS ONE’s style requirements.

2. Thank you for stating the following financial disclosure: [This work was supported by the National Institutes of Health; NIH 2P20 GM109090-06, NIH R15AG063106-01, and NIH NIGMS P20GM152301.].

• Thank you for your feedback. Please update the Funding Statement to the following: “This work was supported by the National Institutes of Health; NIH 2P20 GM109090-06, NIH R15AG063106-01, and NIH NIGMS P20GM152301. The funders had no role in study design, data collection and analysis, decision to publish, or preparation of the manuscript.”

3. Thank you for stating the following in the Acknowledgments Section of your manuscript: [The authors would like to thank Drs. Jaap van Dieën, Mark Grabiner, Alfred Fisher, and Mukul Mukherjee for their thoughtful feedback and manuscript revisions. This work was supported by the National Institutes of Health; NIH 2P20 GM109090-06, NIH R15AG063106-01, and NIH NIGMS P20GM152301.]

Please remove any funding-related text from the manuscript and let us know how you would like to update your Funding Statement. Currently, your Funding Statement reads as follows: [This work was supported by the National Institutes of Health; NIH 2P20 GM109090-06, NIH R15AG063106-01, and NIH NIGMS P20GM152301.].

• Thank you for your feedback.

o We removed funding-related text and revised the Acknowledgments Section to the following “The authors would like to thank Drs. Jaap van Dieën, Mark Grabiner, Alfred Fisher, and Mukul Mukherjee for their thoughtful feedback and manuscript revisions.”.

o Please update the Funding Statement to the following: “This work was supported by the National Institutes of Health; NIH 2P20 GM109090-06, NIH R15AG063106-01, and NIH NIGMS P20GM152301. The funders had no role in study design, data collection and analysis, decision to publish, or preparation of the manuscript.”

4. In the online submission form, you indicated that [The raw data supporting the conclusions of this article will be made available by the authors, without undue reservation.].

• Noted. All unidentified data has been added as supplementary files. Other data that include personal identification numbers and other personal information cannot be shared publicly as it would breach compliance with the protocol approved by the research ethics board.

5. We note that there is identifying data in the Supporting Information file <SupplementaryDataSheet_1_ParticipantsDemographicsHistoryofFalls.xlsx>. Due to the inclusion of these potentially identifying data, we have removed this file from your file inventory. Prior to sharing human research participant data, authors should consult with an ethics committee to ensure data are shared in accordance with participant consent and all applicable local laws.

-Location data

Please remove or anonymize all personal information (ID,AGE,)ensure that the data shared are in accordance with participant consent, and re-upload a fully anonymized data set. Please note that spreadsheet columns with personal information must be removed and not hidden as all hidden columns will appear in the published file.

• Thank you for bringing this to our attention. All demographics and other personal information has been removed.

• Thank you for your feedback. A caption for the Supporting Information has been added to the end of the manuscript and all in-text citations has been revised according to PLOS ONE format.

General comments to the editors and reviewers:

We would like to thank all reviewers for taking the time to provide thoughtful feedback. The following is our response to each of the reviewers’ comments.

Reviewer 1

General Comments

The manuscript presents interesting results that are of interest to biomechanics and wider community as it relates to slip-induced falls. Specifically, the study presents findings that the bilateral slips are more complex in nature and result in higher fall rates compared to the unilateral slips.

We would like to thank you for providing these insightful feedback. We appreciate it.

Authors’ Response to Reviewer 1’s General Comments:

General comments: While the study is really interesting, as a reader/reviewer, I had a very hard time comprehending all the information, due to repeated information/sentences and multivariable analysis with many results presented all together. I suggest to reorganize the results by including subsections and emphasizing the most important results within each of them. In addition, selection of specific timings or events was not well justified. The whole analysis and results comparison was based on those, which requires clarification or standardize representation (and/or potentially rerunning the analysis).

1. While the study is really interesting, as a reader/reviewer, I had a very hard time comprehending all the information, due to repeated information/sentences and multivariable analysis with many results presented all together. I suggest to reorganize the results by including subsections and emphasizing the most important results within each of them.

We revised the results section and added subsections emphasizing the most important findings to reflect a more organized and easier to read outline as you suggested.

2. In addition, selection of specific timings or events was not well justified. The whole analysis and results comparison was based on those, which requires clarification or standardize representation (and/or potentially rerunning the analysis).

The methods section have been revised, and all timings and events have been justified. (lines 275-288).

Authors’ Response to Reviewer 1’s Specific Comments:

Introduction:

1. The last two sentences of the first paragraph in Intro are slightly ambiguous. Are you making comparison of current numbers to those or are you referring to words in their papers. Please rephrase.

These sentences have been revised as recommended. (lines 55-64)

2. Please try to avoid repeating information. for example, shorten last sentence before Section 4 and also spell out angular momentum instead of using L. Note that you defined L as a specific variable. Similarly, see paragraph 2 on page 13 for repeating sentences or definitions.

The manuscript has been revised according to your suggestions and cleared from repetitive information.

Methods:

3. P.7 Line158: Ideally, you should reorganize the Subfigures in Fig. 1 to call this Fig. 1A as it appears first. Simply switch A and B subfigures.

We agree that Fig. 1A should appear first in text. We revised the text to make sure Fig. 1A appears first, as we prefer this subfigure organization. We revised the sentence: “The protocol is outlined in Figure 1.A and all experimental procedures took place in the main Motion Analysis Laboratory (Figure 1.B & E)” (lines 176-178).

4. Why were slips induced during 0-33% of stance phase and not at the heel strike? This does not seem the same as most commonly reported in the literature. Please provide supporting statement/reference from existing literature.

Several previous research studies have identified 0-33% of the stance phase as the load absorption phase using shear ground reaction force data in healthy and diseased populations (Di Gregorio & Vocenas, 2021; Vaverka et al., 2015). The WASP was triggered at heel strike; however, we specifically mention that slips were induced between 0-33% to account for any human delay and error. Because slips were induced manually, such limitation exposes us to human error which has been discussed in the limitations section. However, we agree that all slips were triggered at the heel strike stage which was later confirmed in Visual 3D. Trials that did not result in slip initiation at the heel strike phase were excluded from the analyses. The method section has been revised accordingly (lines 206-211).

5. P8 Line 176: Were during bilateral slips both WASPs activated simultaneously? It seems not, if leading foot was during early and trailing foot was during late stance slip? Please clarify and revise that paragraph. Why was WASP not actuated at the foot touch down? Additionally, how does the instructions of continuing to walk vs stop walking after slip onset effect the end results?

In bilateral slips both WASPs activated simultaneously (i.e., at the same exact timepoint) when the dominant foot entered early stance (Figure 1.F), initiating an early-stance slip on the dominant (leading) foot and a late-stance slip (66.8–100% of stance) on the non-dominant (trailing) foot prior to swing phase. Each WASP was triggered remotely and manually by the same researcher to reduce inter-assessor variability since WASPs were not manufactured with an automated feature that detects gait events during the stance phase. Because all WASPs connect to the same device through Bluetooth and can only be triggered manually and simultaneously by the same researcher, bilateral slips were triggered when the dominant foot reached early stance, which automatically triggered the non-dominant foot’s WASP during late stance. Slips were initiated at the non-dominant’s foot late stance prior to swing phase and the foot continued to slip after touchdown. Participants were instructed to try their best not to fall upon slip initiation as if they were experiencing a slip outdoors until either equilibrium or a fall is reached and then to stop walking. The experimental protocol section of the method has been revised accordingly (lines 212-222).

6. P9. Did the authors check that the gait pattern returned to a normal walking gait after the slip? How did the authors guarantee that the subjects did not alter the gait as to expect more slip perturbations?

All participants were instructed to try their best not to fall upon slip initiation as if they were experiencing a slip outdoors until either equilibrium or a fall is reached and then to stop walking. Upon concluding each trial, researchers walked to each participant and assisted them to carefully walk back to the start line, unhooked the harness for them to sit down and rest on a chair, reset the WASP, and cleaned the floor. Thus, all participants stopped walking as soon as they reached either a fall or a recovery, and no additional movements have been produced to further alter gait or lead to additional slip perturbations. The experimental protocol section of the method has been revised accordingly (lines 222-227).

7. Why was slipping timeline selected as the slip onset. Why it was not measured from the heel strike to have an objective measure and for comparison between all slips that may start at different gait instances? I would suggest to revise that or provide justification.

Slip onset was chosen as the start of the slipping timeline instead of heel strike to normalize all trials to the same slip start and ensure that the analyzed biomechanical data used to quantify slip severity and diversity are data extracted during the actual slip; thus, the slipping timeline has been identified when the dominant foot start sliding until both feet reached a complete stop. The biomechanical analyses section of the methods has been revised accordingly (lines 246-251).

8. Why focusing only within 30 msec of slip initiation or 30msec of trailing limb touchdown? Why is H-reflex latency important for determining foot velocities? Does this has to do anything with friction demands and eliminates effects of human reflexes?

We chose to initially inspect foot velocities within 30 msec of slip initiation and trailing limb touchdown to quantify the velocity of the foot solely due to the reduced frictional demands caused by the slips before any human reflexes interfere, which could lead to increase or decrease in foot velocity. The text has been revised accordingly in order to provide more clarity (lines 257-281).

9. P11 Line 228: It is unclear what the authors are trying to imply here by 'because whole-body angular impulse..' Please explain how is angular impulse used and calculated in this study as a metric.

Angular impulse was not calculated or used in any analyses within this manuscript. Only whole body angular momentum was calculated and used as an outcome measure at three timepoints: 1) slip initiation, 2) trailing limb touchdown, 3) 100ms post trailing limb touchdown. However, upon review, we agree this sentence could be misleading and confusing. The sentence was meant to address the rationale and reason behind choosing foot position and velocity relative to the

---

## [Decision Letter · Decision Letter 1]

9 Jul 2025

Higher Fall Rates and Broader Kinematic Diversity in Bilateral Versus Unilateral Unconstrained Slips

PONE-D-25-09847R1

Dear Dr. Ouattas,

We’re pleased to inform you that your manuscript has been judged scientifically suitable for publication and will be formally accepted for publication once it meets all outstanding technical requirements.

Kind regards,

Anne E. Martin

Academic Editor

PLOS ONE

Additional Editor Comments (optional):

Reviewers' comments:

Reviewer's Responses to Questions

**Comments to the Author**

1. If the authors have adequately addressed your comments raised in a previous round of review and you feel that this manuscript is now acceptable for publication, you may indicate that here to bypass the “Comments to the Author” section, enter your conflict of interest statement in the “Confidential to Editor” section, and submit your "Accept" recommendation.

Reviewer #1: All comments have been addressed

Reviewer #2: All comments have been addressed

2. Is the manuscript technically sound, and do the data support the conclusions?

Reviewer #1: Yes

Reviewer #2: Yes

3. Has the statistical analysis been performed appropriately and rigorously? 

Reviewer #1: Yes

Reviewer #2: Yes

4. Have the authors made all data underlying the findings in their manuscript fully available?

Reviewer #1: Yes

Reviewer #2: Yes

5. Is the manuscript presented in an intelligible fashion and written in standard English?

Reviewer #1: Yes

Reviewer #2: Yes

6. Review Comments to the Author

Reviewer #1: Thanks to the authors for carefully addressing reviewers' comments and providing explanations. There are only two minor comments that authors might consider addressing.

Sentence in the introduction: "Over the past two decades, non-fatal injuries due to slip-induced falls increased by 160% (Berg, Alessio, Millis, & Tong, 69 1997), while fatal injuries increased by 300% (Lambert & Sattin, 1988)..."

The sentence in the introduction about two decades is confusing as the two references provided are approximately 30 years old. It is unclear if they compare old data to the new data or refer to the 30+ year old data itself. Please include some more recent statistics or additional reference if this was meant as a comparisons between the two or replace those with a much more recent ones.

Please replace variable L with the Whole-body angular momentum in the subtitle :"Differences in L across slip types". Same for the figure captions.

Reviewer #2: The authors have done an excellent job to address my comments and suggestions in the previous review. I do not have any further comments and suggestions.

7. PLOS authors have the option to publish the peer review history of their article (what does this mean? ). If published, this will include your full peer review and any attached files.

**Do you want your identity to be public for this peer review?** For information about this choice, including consent withdrawal, please see our Privacy Policy .

Reviewer #1: No

Reviewer #2: No

---

## [Editor Report · Acceptance letter]

PONE-D-25-09847R1

PLOS ONE

Dear Dr. Ouattas,

I'm pleased to inform you that your manuscript has been deemed suitable for publication in PLOS ONE. Congratulations! Your manuscript is now being handed over to our production team.

Kind regards,

on behalf of

Dr. Anne E. Martin

Academic Editor

PLOS ONE